# Efficient Turing Machine Simulation with Transformers

**Qian Li**
Shenzhen International Center For Industrial And Applied Mathematics
Shenzhen Research Institute of Big Data
Shenzhen, China
Email: `liqian.ict@gmail.com`

**Yuyi Wang**
CRRC Zhuzhou Institute & Tengen Intelligence Institute
Zhuzhou, China
Email: `yuyiwang920@gmail.com`

## Abstract

Constant bit-size Transformers are known to be Turing complete, but existing constructions require $\Omega(s(n))$ chain-of-thought (CoT) steps per simulated Turing machine (TM) step, leading to impractical reasoning lengths. In this paper, we significantly reduce this efficiency gap by proving that any $(t(n), s(n))$-bounded multi-tape TM can be simulated by a constant bit-size Transformer with an optimal $O(s(n))$-long context window and only $O(s(n)^c)$ CoT steps per TM step, where $c > 0$ can be made arbitrarily small by letting the Transformers' head-layer product sufficiently large. In addition, our construction shows that sparse attention with fixed geometric offsets suffices for efficient universal computation. Our proof leverages multi-queue TMs as a bridge. The main technical novelty is a more efficient simulation of multi-tape TMs by synchronous multi-queue TMs, improving both time and space complexity under stricter model assumptions.

## 1 Introduction

Transformer-based large language models (LLMs) equipped with chain-of-thought (CoT) reasoning (Gemini et al., 2023; Anthropic, 2024; Guo et al., 2025; OpenAI, 2025) have shown remarkable performance in various challenging reasoning tasks, including solving mathematical problems (Luong & Lockhart, 2025) and generating computer programs (Rein et al., 2023; Alberto Manzi, 2025). Beyond the empirical successes, there has been growing interest in understanding the mechanisms underlying the reasoning abilities of Transformers. A recent line of work (Pérez et al., 2021; Bhattamishra et al., 2020; Merrill & Sabharwal, 2024; Li et al., 2024; Qiu et al., 2024; Yang et al., 2025a; Bavandpour et al., 2025; Li & Wang, 2025) has established the Turing completeness of Transformers. In particular, it turns out (Li & Wang, 2025) that even constant bit-size Transformers, where the number of parameters and precision are both constant independent of the input length, are Turing complete, provided sufficiently long context windows and chain-of-thought (CoT) reasoning steps are allowed. These results demonstrate that Transformers' reasoning ability is universal in principle.

However, critical gaps remain in our theoretical understanding: the efficiency of such simulations is left open. Table 1 summarizes the efficiency of existing Turing completeness constructions. In particular, the constant bit-size construction of Li & Wang (2025) achieves an optimal context window of length $O(s(n))$, which reflects the amount of memory required for reasoning, but incurs a cost of $\Omega(s(n))$ CoT steps to simulate one Turing machine (TM) step. Here $s(n)$ denotes the space bound of the simulated computation. For many problems, this $s(n)$-factor slowdown is prohibitively large, causing the total CoT length to far exceed the number of reasoning steps observed in practice. Therefore, as pointed out by Li & Wang (2025), it is not only of theoretical interest but also of practical significance to investigate whether the slowdown can be avoided without compromising optimality in other aspects.

Table 1: Comparing Transformers' complexity to simulate $(t(n), s(n))$-bounded Turing machines. 'Dim.' denotes embedding dimension, 'Window' means effective window size, and 'Total CoT' denotes the total CoT length. This paper focuses on the nontrivial case where $t(n), s(n) \geq n$.

| Source | Precision | Dim. | Window | Total CoT |
|---|---|---|---|---|
| Pérez et al. (2021) | $O(\log t(n))$ | $O(1)$ | $n + t(n)$ | $O(t(n))$ |
| Bhattamishra et al. (2020) | unbounded | $O(1)$ | $n + t(n)$ | $O(t(n))$ |
| Merrill & Sabharwal (2024) | $O(\log t(n))$ | $O(1)$ | $n + t(n)$ | $O(t(n))$ |
| Li et al. (2024)[*] | $O(1)$ | $O(\log t(n))$ | $O(t(n) \log t(n))$ | $O(t(n) \log t(n))$ |
| Qiu et al. (2024) | $O(\log t(n))$ | $O(1)$ | $O(t(n) \log t(n))$ | $O(t(n) \log t(n))$ |
| Yang et al. (2025a)[†] | $O(\log t(n))$ | $O(1)$ | $s(n)$ | $O(t(n))$ |
| Bavandpour et al. (2025) | $O(\log t(n))$ | $O(1)$ | $n + t(n)$ | $O(t(n))$ |
| Li & Wang (2025) | $O(1)$ | $O(1)$ | $s(n)$ | $O(t(n) \cdot s(n))$[¶] |
| **This work** | $O(1)$ | $O(1)$ | $s(n)$ | $O(t(n) \cdot s^c(n))$[‡] |

[*] Any multi-tape TM running in time $t(n)$ can be simulated by a Boolean circuit of size $O(t(n) \log t(n))$ (Arora & Barak, 2009).
[†] The construction incorporates a reduction mechanism to recursively erase intermediate CoT steps.
[¶] By a standard technique, a $t$-time $s$-space multi-tape TM can be converted into a Post machine running in time $O(t \cdot s)$ and space $O(s)$.
[‡] The exponent $c > 0$ can be made arbitrarily small by letting the precision and embedding size sufficiently large constants.

## 1.1 OUR CONTRIBUTION

We make significant progress on the open efficiency problem by reducing the per-step slowdown from $s(n)$ to $s(n)^c$, where the exponent $c > 0$ can be made arbitrarily small. Formally, letting the *head-layer product* denote the product of the number of attention heads and the number of layers, our main theorem is stated as follows:

**Theorem 1.** *Any $(t(n), s(n))$ time-space bounded $k$-tape Turing Machine can be simulated by a constant bit-size Transformer with head-layer product $K$ and context window length $s(n) + 1$, that takes $O(s(n)^{6k/K})$ CoT steps for each simulated TM step, leading to a total CoT length of $O(t(n) \cdot s(n)^{6k/K})$.*

Moreover, our construction exhibits a sparse *geometric-offset attention* property: every query attends to only a few tokens located at fixed relative positions, with offsets chosen in a geometric progression[1]. Interestingly, the geometric-offset attention pattern is in the same spirit as attention behavior observed: empirical analyses report that attention is typically sparse and spatially clustered, with many heads focusing on local neighborhoods and a minority capturing structured long-range connections, e.g. (Xiao et al., 2024; Ribar et al., 2024; Jiang et al., 2024; OpenAI, 2025). As a consequence, each token can be produced in $O(1)$ time, and simulating Turing machines incurs only a per-step slowdown of $O(s(n))^c$, in contrast to the $t(n)$ overhead in full attention. This suggests that the wide concern that the quadratic time complexity of full attention constitutes a fundamental throughput bottleneck for long-context Transformers (e.g., (Khan et al., 2022; Sarikaya, 2025))) is not necessarily a principled expressiveness limitation of Transformer-based architectures.

Our results highlight geometric-offset attention as a promising architectural direction and principled design choice for efficient reasoning: a query attends to only the tokens $1, 2, 4, 8, \ldots, 2^i, \ldots$ steps earlier [2]. Such exponentially spaced connections avoids the quadratic overhead of full attention, while still being sufficient for efficient universal computation as shown by our theory. Notably, similar exponentially spaced or logarithmically sparse patterns have already been explored in practice, such as in LogSparse Transformers (Li et al., 2019) and PowerAttention (Chen et al., 2025). Our

---

[1] Specifically, in Theorem 1, the offsets are $\lceil s(n)^{1/k'} \rceil, \lceil s(n)^{1/k'} \rceil^2, \ldots, \lceil s(n)^{1/k'} \rceil^{k'}$ with $k' = K/(6k)$.
[2] In the regime of practical interest, the head-layer product $K$ is typically large (e.g. $K$ on the order of $10^3$-$10^4$ in modern LLMs). Moreover, it is natural to focus on $k = 2$, since 2-tape TM can efficiently simulate multi-tape TMs with only a logarithmic slowdown. So, if we take $K = 6 \times 10^3$ and $k = 2$, then the common ratio would be $\lceil s(n)^{6k/K} \rceil = 2$ for any $s(n) \leq 2^{500}$.

results thus provide a theoretical evidence that these sparse designs can retain the computational universality while improving efficiency.

Our proof strategy is inspired by the approach of Li & Wang (2025), which established a bridge between TMs and Transformers via *one-queue* Turing machines (a.k.a. Post machines). We generalize this idea by introducing *multi-queue* Turing machines as the bridge. Concretely, Step 2 of our proof can be seen as a generalization of that of Li & Wang (2025) to multi-queue machines; the main technical novelty of this work lies in Step 1.

**Step 1: From multi-tape TMs to multi-queue TMs.** Previous works have studied the relationship between multi-tape and multi-queue machines; see (Petersen, 2013) for an overview of existing results. In particular, Hühne (1993) showed that any $(t(n), s(n))$-bounded $k$-tape TM can be simulated by a $k'$-queue TM that runs in $\left(O(t(n)^{1+1/k'}), O(t(n)^{1+1/k'})\right)$ time and space. However, their multi-queue model is more permissive: queues are allowed to remain idle in a step (i.e., neither pop nor push symbols). By contrast, we consider the more restrictive *synchronous* model, where every queue pops exactly one symbol and also appends exactly one symbol at each step. Despite this restriction, we show that any $(t(n), s(n))$-bounded $k$-tape TM can be simulated by a synchronous $(6kk')$-queue TM that runs in $\left(O(t(n) \cdot s(n)^{1/k'}), O(s(n))\right)$ time and space (Theorem 2). Our improvement over (Hühne, 1993) is threefold: (i) extending the result to the more restrictive synchronous model, (ii) reducing the space complexity from $t(n)^{1+1/k'}$ to $s(n)$, and (iii) reducing the time slowdown from $t(n)^{1/k'}$ to $s(n)^{1/k'}$.

**Step 2: From multi-queue TMs to Transformers.** Once Step 1 is established, we adapt the simulation technique of Li & Wang (2025) to synchronous multi-queue TMs. Specifically, we prove that any synchronous $(t(n), s(n))$-bounded $K$-queue TM can be simulated by a constant bit-size Transformer with head-layer product $K$, context window $O(s(n))$, and CoT length $O(t(n))$ (Theorem 3). Now, our main theorem, namely Theorem 1, follows immediately from these two steps.

## 1.2 Other related works

**Formal-language expressivity.** A substantial body of work (Hahn, 2020; Yao et al., 2021; Hao et al., 2022; Feng et al., 2023; Chiang et al., 2023; Merrill & Sabharwal, 2023; Yang et al., 2024; Barcelo et al., 2024; Merrill & Sabharwal, 2025; Chen et al., 2024a) studies Transformers through the lens of formal-language recognition, asking which language classes are captured under different architectural and positional-encoding assumptions, e.g., absolute vs. relative PE, with vs. without CoT, restricted attention, precision assumptions, and depth. The survey by Strobl et al. (2024) provides a comprehensive overview of upper and lower bounds across variants, clarifying how design choices affect expressivity.

**Reasoning efficiency.** Current reasoning models tends to overthink, generating excessively long CoTs that causes significant computational redundancy (Chen et al., 2024b; Sui et al., 2025). Moreover, it has been shown that longer CoTs do not necessarily improve accuracy and can even hurt it (Wu et al., 2025; Yang et al., 2025b; Jin et al., 2024). To mitigate overthinking, proposed methods include RL with length penalties (Luo et al., 2025; Aggarwal & Welleck, 2025), SFT on variable-length traces (Ma et al., 2025; Xia et al., 2025), and latent reasoning that avoids generating explicit CoTs (Hao et al., 2024; Su et al., 2025). Beyond shortening CoTs, another line of work seeks to maintain strong reasoning with smaller models by applying compression techniques (e.g., distillation, quantization, and pruning) and by directly training smaller models with RL (Li et al., 2023; 2025; Zhu et al., 2024). For comprehensive surveys of efficient reasoning methods, see (Feng et al., 2025) and (Sui et al., 2025).

## 2 Preliminaries

Throughout this paper, we adopt the following standard notation. The sets of real and natural numbers are denoted by $\mathbb{R}$ and $\mathbb{N}$ respectively, and $[n] := 1, 2, \ldots, n$ for $n \in \mathbb{N}$. Vectors and sequences are written in bold lowercase (e.g., $\boldsymbol{x}$), and matrices in bold uppercase (e.g., $\boldsymbol{A}$). For a vector or

sequence $\boldsymbol{x}$, let $x_i$ denote its $i$-th element, and let $\boldsymbol{x}_{j:i}$ denote $(x_j, x_{j+1}, \cdots, x_i)$ for $j \leq i$. For a matrix $\boldsymbol{A}$, let $A_{i,j}$ denote the $j$-th element in the $i$-th row. We write $\boldsymbol{0}_n$ for the $n$-dimensional zero vector.

## 2.1 MULTI-TAPE TURING MACHINES

A $k$-tape Turing machine (TM) has $k$ tapes, one of which initially contains the input and is called the input tape. Each tape is infinite to the right and bounded on the left, and equipped with a tape head. It can be defined as a tuple $\langle \Sigma, Q, \delta \rangle$ where

- $\Sigma$ is a finite tape alphabet, including 0, 1, and a blank symbol $\perp$.
- $Q$ is a finite set of states, including a start state $q_{start}$ and a halting state $q_{halt}$.
- $\delta : Q \times \Sigma^k \to Q \times \Sigma^{k-1} \times (\{\text{Left}, \text{Stay}, \text{Right}\})^k$ is a transition function.

Initially, the input tape contains a finite $\{0, 1\}$-string $x$ as the input, and the other $k - 1$ tapes are all empty. The computation repeats the following until the TM enters $q_{halt}$: if the machine is in state $q \in Q$ and reading symbols $(\sigma_0, \cdots, \sigma_{k-1})$ from the tapes, and if $\delta(q, \sigma_0, \cdots, \sigma_{k-1}) = (q', z_0, \sigma'_1, z_1, \cdots, \sigma'_{k-1}, z_{k-1})$ where $z_i \in \{\text{Left}, \text{Stay}, \text{Right}\}$, then the TM changes its state to $q'$, writes $\sigma'$ to the working tapes, and moves the heads according $z$.

Without loss of generality, and for technical convenience, we assume that at the start of the computation the Turing machine first performs a scan over the input tape.

The *space complexity* of a $k$-tape Turing machine is the function $s(n)$ that, for each input length $n$, gives the maximum number of distinct cells visited on the $k$ tapes during the computation on any input of length $n$.

## 2.2 MULTI-QUEUE TURING MACHINES

In contrast to multi-tape TMs that operate on tapes, multi-queue TMs operate on queues. Previous works on multi-queue TMs (e.g. (Hühne, 1993)) studied the model in which each queue can remain idle in a step (i.e., neither pop nor push symbols), or equivalently (up to a constant factor slowdown), only one queue can pop or append an element at each step. In this paper, for technical reasons, we consider a more restrictive variant, called the *synchronous multi-queue TM*, where each queue pops exactly one element and also append exactly one element at each step. Note that a synchronous multi-queue TM can be simulated by the traditional model with only a constant factor slowdown, whereas the reverse direction is unlikely to hold.

Formally, a $k$-queue synchronous TM has $k$ queues, one of which initially contains the input and is called the input queue. It can be defined as a tuple $\langle \Sigma, Q, \delta \rangle$ where

- $\Sigma$ is a finite tape alphabet, including 0, 1, and a blank symbol $\perp$.
- $Q$ is a finite set of states, including a start state $q_{start}$ and a halting state $q_{halt}$.
- $\delta : Q \times \Sigma^k \to Q \times \Sigma^k$ is a transition function.

Initially, the input queue contains the input string, and the other $k - 1$ queues contains only blank symbols. The computation repeats the following until the TM enters $q_{halt}$: if the machine is in some state $q \in Q$ and reading symbols $(\sigma_0, \cdots, \sigma_{k-1})$ popped from the queues, and if $\delta(q, \sigma_0, \cdots, \sigma_{k-1}) = (q', \sigma'_0, \cdots, \sigma'_{k-1})$, then the TM changes its state to $q'$ and appends $\sigma'$ to the queues.

The *space complexity* of a $k$-queue (synchronous) TM is the function $s(n)$ that, for each input length $n$, gives the maximum total length of the $k$ queues during the computation on any input of length $n$.

## 2.3 TRANSFORMER ARCHITECTURE

Let $\mathcal{V}$ be the vocabulary. A decoder-only Transformer $\mathsf{TF}_\theta$ is a parameterized function $\bigcup_{i \geq 1} \mathcal{V}^i \to \mathcal{V}$ composed of an embedding layer, multiple decoder layers, and an output layer.

**Token embedding layer**  For each previous token $v_j \in \mathcal{V}$ (with $j \leq i$), we map it to a $d$-dimensional embedding vector $\boldsymbol{h}_j^0 := \mathsf{emb}(v_j)$. We do *not* add absolute positional embeddings to $\boldsymbol{h}_j^0$. Instead, we introduce relative positional information only inside the self-attention scores, following the formulation of Shaw et al. (2018). Here, we call $d$ the *model dimension* or *embedding dimension*.

**Decoder Layers**  The $\boldsymbol{h}_j^0$ is then processed by $L$ decoder layers. Each decoder layer $\mathsf{dec}_\ell$ consists of a self-attention sub-layer followed a feed-forward network sub-layer, with residual connections and layer normalization applied around each sub-layer. For simplicity and following (Li et al., 2024), we omit layer normalization; see (Li et al., 2024) for how it can be incorporated without affecting our results. In addition, following common practice[3], we use $\mathrm{hardmax}$ as a proxy for $\mathrm{softmax}$.

- Self-attention sublayer: For each head $k = 0, 1, \cdots, H-1$, compute the attention score as[4]

$$\boldsymbol{s}_{k,i}^\ell = \mathsf{hardmax}\left(\langle \boldsymbol{h}_1^\ell \cdot \boldsymbol{K}_k^\ell + \mathsf{pos}(i-1), \boldsymbol{h}_i^\ell \cdot \boldsymbol{Q}_k^\ell\rangle, \cdots, \langle \boldsymbol{h}_i^\ell \cdot \boldsymbol{K}_k^\ell + \mathsf{pos}(i-i), \boldsymbol{h}_i^\ell \cdot \boldsymbol{Q}_k^\ell\rangle\right),$$

  where $\mathsf{pos}(i-j) \in \mathbb{R}^{d/H}$. The head output is then

$$\boldsymbol{a}_{i,k}^\ell = \sum_{j=1}^i s_{k,i,j}^\ell \cdot v_k^\ell(\boldsymbol{h}_i^\ell), \text{ where } v_k^\ell(h) := \boldsymbol{h} \cdot \boldsymbol{V}_k^\ell$$

  Concatenating all heads yields $\boldsymbol{a}_i^\ell = \left((\boldsymbol{a}_{i,1}^\ell)^T, \cdots, (\boldsymbol{a}_{i,H}^\ell)^T\right)^T \in \mathbb{R}^d$. The residual update is

$$\boldsymbol{h}_i^{\ell+0.5} := \boldsymbol{W}^\ell \cdot \boldsymbol{a}_i^\ell + \boldsymbol{b}^\ell + \boldsymbol{h}_i^\ell,$$

  Here $\boldsymbol{Q}_k^\ell, \boldsymbol{K}_k^\ell, \boldsymbol{V}_k^\ell \in \mathbb{R}^{d \times d/H}, \boldsymbol{W}^\ell \in \mathbb{R}^{d \times d}$ and $\boldsymbol{b}^\ell \in \mathbb{R}^d$ are learnable parameters.

- Feed-forward sub-layer: Apply a fully-connected $\mathrm{ReLU}$ neural network $\mathsf{FF}^\ell$:

$$\boldsymbol{h}_i^{\ell+1} = \mathsf{FF}^\ell(\boldsymbol{h}_i^{\ell+0.5}) + \boldsymbol{h}_i^{\ell+0.5}.$$

**Output layer**  The final representations $\boldsymbol{h}_i^L$ is mapped to the vocabulary by:

$$v_{i+1} := \mathsf{out}(\boldsymbol{h}_i^L) = \arg\max(\boldsymbol{W}^{\mathsf{out}} \cdot \boldsymbol{h}_i^L + \boldsymbol{b}^{\mathsf{out}}), \text{ where } \boldsymbol{W}^{\mathsf{out}} \in \mathbb{R}^{|\mathcal{V}| \times d} \text{ and } \boldsymbol{b}^{\mathsf{out}} \in \mathbb{R}^{|\mathcal{V}|}.$$

A Transformer is said to have a context window of length $s$ if the query can attend only to the most recent $s$ tokens. Its *bit-size* is defined as $p \times |\boldsymbol{\theta}|$, where $p$ denotes the precision, i.e., all learnable parameters are stored with $p$ bits per scalar, and $|\boldsymbol{\theta}|$ the number of learnable parameters. The *head-layer product* is defined as the product of the number of heads $H$ and the number of layers $L$.

## 3 STEP I: EFFICIENT SIMULATION OF MULTI-TAPE TMS BY MULTI-QUEUE TMS

This section proves the following theorem.

**Theorem 2.** *Any $(t(n), s(n))$ time-space bounded $k$-tape Turing Machine $M$ can be simulated by a synchronous $6kk'$-queue Turing Machine $M'$ which is $(O(t(n) \cdot s(n)^{1/k'}), O(k \cdot s(n)))$ time-space bounded.*

The basic idea is to simulate each tape using two stacks, and then simulate a single stack with $k'$ levels of queues whose sizes grow geometrically. Concretely, level $i$ consists of a *content queue* of size $2\lceil s^{1/k'}\rceil^i$ (realized as two *half queues* of size $\lceil s^{1/k'}\rceil^i$ each), and an *auxiliary buffer queue* of size $2\lceil s^{1/k'}\rceil^i$. The content queue maintains the invariant that: a block of real symbols followed by a block of dummy symbols. The top of the simulated stack always sits at the boundary of real and dummy cells in level 1, while deeper stack elements are stored in higher levels.

---

[3]As we will see, in our construction, the input to $\mathrm{hardmax}$ is always a one-hot vector, and so is the output. As a consequence, our Transformer construction in Theorem 3 works with all variants of hardmax

[4]Our conclusions (specifically Theorem 3 and Theorem 1) continue to hold under other relative positional encoding formations, e.g. using $\langle \boldsymbol{h}_j^\ell \cdot \boldsymbol{K}_k^\ell, \boldsymbol{h}_i^\ell \cdot \boldsymbol{Q}_k^\ell\rangle + \mathsf{pos}(i-j)$ instead of $\langle \boldsymbol{h}_j^\ell \cdot \boldsymbol{K}_k^\ell + \mathsf{pos}(i-j), \boldsymbol{h}_i^\ell \cdot \boldsymbol{Q}_k^\ell\rangle$. In fact, our construction in Theorem 3 uses relative positional encoding to realize attention to tokens located at fixed relative positions.

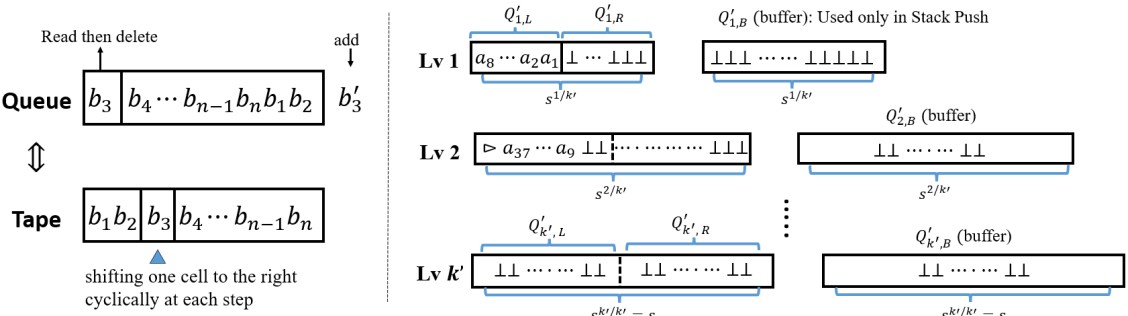

Figure 1: Left: A queue can be viewed as a tape with the head shifting one cell to the right cyclically at each step. Right: Illustration of the queues to simulate a stack.

- **Stack Push.** To push a new symbol onto the stack, we insert it into the first dummy cell of level 1. If level 1 becomes full, half of its content is moved to level 2; if level 2 is full, half of its content is moved to level 3, and so on.

- **Stack Pop.** To pop from the stack, we remove the last real symbol from level 1. If level 1 becomes empty, refill the left half queue by pulling elements from level 2, again recursively if needed.

The crucial property is that higher-level queues are much larger but accessed far less frequently, so expensive transfers are rare. A basic push or pop costs $O(s^{1/k'})$. A transfer between levels $i-1$ and $i$ costs $O(s^{i/k'})$ but occurs only once every $O(s^{(i-1)/k'})$ simulated stack steps, giving an amortized overhead of $O(s^{1/k'})$ per stack step. Thus simulating $t(n)$ steps taks $O(t(n) \cdot s^{1/k'})$ time while using overall queue space $O(s)$. Since each tape equals two stacks and each stack requires $3k'$ queues, simulating $k$ tapes needs $6kk'$ queues in total.

*Proof.* Since each tape can be simulated by two stacks, it suffices to describe how to simulate a single stack PD of the TM $M$ using $3k'$ queues of $M'$:

$$(Q'_{1,L}, Q'_{1,R}, Q'_{1,B}), \cdots, (Q'_{i,L}, Q'_{i,R}, Q'_{i,B}) \cdots, (Q'_{k',L}, Q'_{k',R}, Q'_{k',B}).$$

Here, the subscripts "L", "R", and "B" stand for "Left Half", "Right Half", and "Buffer", whose precise roles will be clarified below. We assume that PD contains a distinguished bottom element that is never popped. During the simulation, the length of $Q'_{i,L}$ and $Q'_{i,R}$ is fixed at $(\lceil s^{1/k'} \rceil)^i$, and $|Q'_{i,B}|$ is fixed at $2(\lceil s^{1/k'} \rceil)^i$. So the total queue length is

$$\sum_{i=1}^{k'} 4 \times (\lceil s^{1/k'} \rceil)^i = O(s).$$

It would be helpful to imagine a queue $Q'_{i,*}$ as a tape $T'_{i,*}$ of the same length, with the tape head shifting exactly one cell to the right cyclically at each step of $M'$.

- Let $\Sigma$ denote the alphabet of PD. Then we define $\Sigma' := \{a, \hat{a}, \tilde{a} \mid a \in \Sigma\} \cup \{\bot\}$ as the alphabet of each $T'_{i,*}$. Here, $\hat{\cdot}$ and $\tilde{\cdot}$ mark the leftmost and rightmost cells of $T'_{i,*}$ respectively, and $\bot$ denotes a dummy placeholder. A tape $T'_{i,*}$ is said to be *full* if it contains only *real symbols* (i.e., non-dummy symbols), and *empty* if it contains only dummies.

- Let $T'_i := T'_{i,L} \circ T'_{i,R}$ be the concatenation of $T'_{i,L}$ and $T'_{i,R}$. Intuitively, $T'_i$ plays the role of the *content queue* of level $i$ in the proof intuition: it stores the actual stack symbols, and its content is arranged as a block of real symbols followed by a block of dummies (from left to right) during the simulation. We call $T'_i$ *balanced* if it contains equal numbers of non-dummies and dummies. Equivalently, this means $T'_{i,L}$ is full and $T'_{i,R}$ is empty.

---

**Algorithm 1** Simulation of pushing $a$ into PD

---

1: Rotate the tape heads until reach the first dummy element of $T_1'$;
2: replace $\perp$ with $a$ in the current cell of $T_1'$;
3: **if** the current cell of $T_1'$ has an accent $\tilde{\cdot}$ **then**
4:     run PUSH(2);
5: **end if**

6: **procedure** PUSH(i)
7:     Rotate the tape heads until reach the first dummy element of $T_i'$;
8:     **while** the current cell of $T_{i-1,R}'$ is non-dummy **do**
9:         write this non-dummy into the current cell of $T_i'$;
10:         write $\perp$ into the current cell of $T_{i-1,R}'$;
11:         rotate the tape heads one cell to the right cyclically.
12:     **end while**
13:     **if** $T_i'$ is full **then**
14:         run PUSH(i+1);
15:     **end if**
16: **end procedure**

---

**Algorithm 2** Simulation of poping a element from PD

---

1: Rotate the tape heads until reach the last non-dummy element of $T_1'$, denoted as $a$;
2: replace $a$ with $\perp$ in the current cell of $T_1'$;
3: **if** the current cell of $T_1'$ has an accent $\hat{\cdot}$ **then**
4:     run procedure POP(2);
5: **end if**
6: **return** a;

7: **procedure** POP(i)
8:     **if** $T_i'$ is not empty **then**
9:         Rotate the tape heads until reach the leftmost element of $T_i'$;
10:         **while** the current cell of $T_i'$ is non-dummy **do**
11:             **if** the current cell of $T_{i-1,L}'$ is non-dummy **then**
12:                 write this non-dummy of $T_{i-1,L}'$ into the current cell of $T_{i,B}'$;
13:             **end if**
14:             write this non-dummy of $T_i'$ into the current cell of $T_{i-1,L}'$;
15:             rotate the tape heads one cell to the right cyclically;
16:         **end while**
17:     **end if**
18:     **if** $T_i'$ is empty **then**
19:         run POP(i+1);
20:     **end if**
21: **end procedure**

---

- During the simulation, we will maintain the following property: The top stack elements of PD are stored in $T_1'$ and deeper elements are stored in $T_2', T_3', \cdots, T_{k'}'$, such that the concatenation $T_{k'}' \circ \cdots \circ T_1' = \mathsf{PD}$.

- By the synchronous condition, at each step of $M'$, every queue $Q_{i,*}'$ pops exactly one symbol and appends exactly one symbol. Accordingly, the head of each tape $T_{i,*}'$ shifts exactly one cell to the right cyclically.

At the beginning of the simulation, PD contains only the distinguished bottom element; all $T_{i,*}'$ are empty except $T_{1,L}'$ stores the bottom element in its leftmost cell; all heads start at the leftmost tape cell. Now, we show how to simulate the stack push and pop operations.

**Stack Push** To push a symbol $a$ on stack PD, $M'$ rotates the tape heads until the first dummy cell of $T_i'$ is reached and replaces it with $a$.

If $T_1'$ becomes full, then the first $|T_1'|/2$ non-dummies of $T_1'$ are pushed into the first $|T_1'|/2$ dummy cells of $T_2'$ (i.e., PUSH(2) in Algorithm 1). Specifically, in PUSH(2), the entire content of $T_{1,L}'$ is copied into these cells of $T_2'$, $T_{1,L}'$ is cleaned, and then the remaining contents of $T_1'$ is shifted left by $|T_1'|/2$ cells (or equivalently, just swap the roles of $T_{1,L}'$ and $T_{1,R}'$). This preserves the arrangement that dummies to the left and dummies to the right. If $T_2'$ is full, then execute PUSH(3), and so on.

**Fact 1.** *Right after PUSH($i$), all $T_1', \cdots, T_{i-1}'$ are balanced.*

**Stack Pop** To pop the top element of PD, $M'$ rotates the head of $T_1'$ to the last non-dummy element[5], outputs it, and replaces it with $\bot$.

If $T_1'$ becomes empty, then the last $|T_1'|/2$ non-dummy elements of $T_2'$ are transferred into the first $|T_1'|/2$ cells of $T_1'$ (i.e., POP(2) in Algorithm 2). Specifically, in POP(2), the last $|T_1'|/2$ non-dummy elements of $T_2'$ are copied into $T_{1,L}'$, after which those cells of $T_2'$ are cleaned. The main challenge to implement POP(2) is that we cannot directly identify the last $|T_1'|/2$ non-dummies of $T_2'$. To overcome this difficulty, we employ the buffer queue $T_{2,B}'$ to temporarily store a block of elements of $T_2'$ during the scan. The intuition is as follows: we scan $T_2'$ from left to right, and move each encountered symbol to $T_{1,L}'$; whenever a cell of $T_{1,L}'$ is about to be overwritten, its previous content is diverted to $T_{2,B}'$. In this way, we maintain the invariant that $T_{2,B'} \circ T_{1,L}' \circ T_2'$ always equals the original content of $T_2'$. In particular, when we reach the first dummy of $T_2'$, we have $T_{2,B'} \circ T_{1,L}'$ equals the original content of $T_2'$. Formally,

- First, rotate the head of $T_2'$ to the leftmost cell; (one can see that the $T_{1,L}'$ and $T_{2,B}'$ heads also locates at the leftmost cell at this time, since $|T_{1,L}'|$ divides $|T_2'|$ and $|T_{2,B}'| = |T_2|$.)

- As the head of $T_2'$ goes from left to right until reaches the first $\bot$,
    1. If the currect cell of $T_{1,L}'$ store a non-dummy, then move this element to $T_{2,B}'$;
    2. The current element of $T_2'$ is copied to the currect cell of $T_{1,L}'$, and then cleaned.

- Finally, the content in $T_{2,B}'$ is copied to $T_2'$, and then cleaned.

If $T_2'$ becomes empty, the procedure recurs to POP(3) and so on.

**Fact 2.** *Right after POP($i$), all of $T_1', \cdots, T_{i-1}'$ are balanced.*

What remains is to analyze the correctness and efficiency of this simulation.

**Claim 1.** *During the simulation, we always have the following properties.*

(a) *The numbers of dummies and non-dummies of $T_i'$ are both multiples of $|T_{i-1,L}'|$, for any $i = 2, \cdots, k'$.*

(b) *When the head of $T_i'$ is on the first dummy cell, the head of $T_{i-1,L}'$ is on the leftmost cell. Similarly, when the head of $T_i'$ is on the last non-dummy cell, the head of $T_{i-1,L}'$ is on the rightmost cell.*

(c) *At the beginning of the stack push simulation, each $T_i'$ contains at least $|T_{i-1,L}'|$ dummies. Consequently, combining with (a) and (b), it implies that PUSH($i$) is doable whenever it is called.*

(d) *If $T_i'$ is empty, then all higher levels $T_{i+1}', \cdots, T_{k'}'$ are empty as well.*

---

[5]A non-dummy cell is the last one iff it carries the rightmost mark $\widetilde{\phantom{x}}$ or its next cell is dummy. To avoid explictly pre-reading the next cell, we apply a one-step *delayed append* trick: initially, pop the leftmost queue symbol and store it in the finite-state controller $Q$, without appending. Thereafter, at each step (i) pop the next leftmost symbol, and (ii) append a symbol determined by the two most recently popped symbols. In this way, the appended symbol can depend on the leftmost two queue symbols, without explicit pre-reading.

*(e)* *The concatenation of non-dummy contents of $T'_{k'}, \cdots, T'_1$ (in this order) equals exactly the content of* PD *(we start with the bottom element).*

Consequently, $T'_1, \cdots, T'_{k'}$ faithfully simulate the stack PD. *Precisely, whenever* PD *pops the top element,* $M'$ *can obtain it as well.*

*Proof. (a).* Intially, $T'_i$ have 0 non-dummies and $2(\lceil s^{1/k'} \rceil)^i = 2\lceil s^{1/k'} \rceil \cdot |T'_{i-1,L}|$ dummies. The numbers changes only if PUSH(i) or POP(i) is executed. Through PUSH(i), $|T'_{i-1,L}|$ dummies are changed to non-dummies, or $T'_i$ becomes balanced. Through POP(i), $|T'_{i-1,L}|$ non-dummies are changed to dummies, or $T'_i$ becomes balanced.

*(b).* Immediate from (a).

*(c).* At the start of a stack push, $T'_i$ cannot be full, since otherwise PUSH($i+1$) would be executed in the previous execution of PUSH($i$) and would balance $T'_i$. Combining with (a) yields (c).

*(d).* It suffices to show that if $T'_i$ is empty then $T'_{i+1}$ is empty. Suppose $T'_i$ is empty at a time, then either

- $T'_i$ has been empty since the beginning of the simulation, where $T'_{i+1}, \cdots, T'_{k'}$ have been empty as well; or

- $T'_i$ becomes empty after POP($i$). Right after this POP($i$), $T'_{i+1}$ should be empty as well since otherwise POP($i+1$) would be executed which makes $T'_i$ balanced.

*(e).* From the description of POP($i$), one can see that right after POP($i$), the concatenation of non-dummy contents of $T'_i$ and $T'_{i-1,L}$ (in this order) equals the non-dummy content of $T'_i$ before POP($i$). Now, (e) is immediate. $\square$

In the following, we analyze the computational time cost. The simulation of one step (without calling PUSH or POP) costs $O(|T'_1|) = O(s^{1/k'})$. An execution of PUSH($i$) or POP($i$) (without calling PUSH($i+1$) or POP($i+1$)) costs $O(|T'_i|) = O(s^{i/k'})$. Since after PUSH($i$) or POP($i$), all levels $T'_1, \cdots, T'_{i-1}$ are balanced, one can see that the number of stack operations between successive calls to PUSH($i$) or POP($i$) is at least as $|T'_{i-1}|/2 = \Omega(s^{(i-1)/k'})$. So the total simulation time is bounded by

$$t \cdot O(|T'_1|) + \sum_{i=1}^{k'} O(|T'_i|) \cdot \frac{t}{\Omega(s^{(i-1)/k'})} = O(t \cdot s^{1/k'}).$$

Finally, the total space used by the multi-queue machine is

$$2k \cdot \sum_{i=1}^{k'} 4 \times \lceil s(n) \rceil^{i/k'} \le 8k \cdot \sum_{i=1}^{k'} (s(n) + 1)^{i/k'} \le 8k \cdot 2(s(n) + 1) = O(ks(n)). \qquad \square$$

**Remark 1.** *Our proof builds on the high-level idea of Theorem 3.2 in (Hühne, 1993), namely the use of queues of geometrically increasing size. However, their model allows each queue to remain idle in a step (i.e., neither pop nor push symbols), whereas we impose the stricter* synchronous condition *that every queue must pop and append exactly one symbol at every step. This requirement significantly complicates the simulation: when transferring data between adjacent levels, the two participating heads must arrive at the correct cells simultaneously. To resolve the head-alignment issue, our simulation departs from that of Hühne (1993) in a crucial way, including the use of auxiliary queues as buffers and the splitting of each content queue into two halves to regulate head positions.*

## 4 STEP II: EFFICIENT SIMULATION OF MULTI-QUEUE TMS BY TRANSFORMERS

This section proves the following theorem, which together with Theorem 2 immediately implies Theorem 1. Theorem 3 is a generalization of Theorem 4 in (Li & Wang, 2025), extending the single-queue construction to the multi-queue setting.

**Theorem 3.** *Let* TM *be a synchronous $K$-queue Turing machine that, on input $x \in \{0,1\}^n$, uses at most $s_{\max}(n)$ maximum queue length and runs for at most $t(n)$ steps. There exists a constant bit-size Transformer with (i) a head-layer product $K$ and (ii) a context window of length $O(s_{\max}(n))$ that, on input $x$, takes $O(t(n))$ CoT steps and then outputs* TM$(x)$.

The complete proof is given in Appendix B; below we sketch the main idea. Let TM be such a synchronous $K$-queue Turing machine. Because each of the $K$ queues pops exactly one element and also appends exactly one element at each step, their sizes are fixed during the execution of TM. Let $s_0(n), s_1(n), \cdots, s_{K-1}(n)$ denote the queue sizes. Let $s_{\max}(n) \geq \max_r s_r(n)$.

We now construct a constant bit-size Transformer TF with a context window of length $s_{\max}(n)$ to faithfully simulate TM step by step. The intuition is as follows:

- We view each queue as a tape that is (i) infinite to the right and bounded on the left, and (ii) equipped with two tape heads, namely the front head and the rear head. The queue content corresponds to the tape segment between the two heads. Initially, the rear head of the $r$-th tape is positioned at cell $n$, while the front head is at $n - s_r(n) + 1$[6]. At each step, the front head first reads the current symbol, both heads move exactly one cell to the right, and then the rear head writes a symbol.

- We represent the $K$ tapes by stacking them cellwise: the $i$-th token in the Transformer's context records the $i$-th cells of all the $K$ tapes, with the cells pointed by the rear heads corresponding to the newest token. Since the context window length is $s_{\max}(n) = \max_r s_r(n)$, it can store the entire content of the queues. In addition, we set the vocabulary of the Transformer to be $\mathcal{V} = \Sigma^K \times Q$, so that a token can also track the TM state information.

- The transition function of TM takes as input the leftmost elements of the queues together with the current state, and outputs the next state and symbols. To simulate this, we partition the $K$ queues into $L$ groups of size $H := K/L$. In each decoder layer $\ell = 0, \cdots, L-1$, each of the $H$ attention heads retrieves the leftmost symbol of one queue in the $\ell$-th group from previous tokens. In addition, the current state can be retrieved from the current token. Subsequently, a feed-forward network is applied to implement the transition function $\delta$.

**Remark 2.** *Following Li & Wang (2025), we employ unlearnable relative positional encoding that evolves with $n$ as defined in Equation (2). To simulate the given TM on longer inputs, we only need to adjust the relative PE according to Equation (2); all learnable parameters, including their precision, remain unchanged. Thus, for a given TM, there is a single learnable parameter vector that is reused for all input lengths.*

*Our construction is of constant bit-size, since the notion of bit-size refers to the number and precision of* learnable parameters*, and the positional encoding is treated as part of the model architecture rather than as learnable parameters.*

## 5 CONCLUSION AND FUTURE DIRECTIONS

This paper proves that constant bit-size Transformers can simulate multi-tape TMs with an optimal $O(s(n))$ context window and only $O(s(n)^c)$ CoT steps per simulated step, where $c > 0$ can be made arbitrarily small. Beyond the theoretical advance, our construction suggests geometric-offset attention as a promising architectural direction to enhance reasoning efficiency: each CoT token can be generated in $O(1)$ time, demonstrating that dense quadratic attention is not essential for efficient universality. The key technical contribution is a more efficient simulation of multi-tape TMs via multi-queue TMs. We propose three future directions: (i) Can the per-step overhead be further reduced to $O(1)$? (ii) Our construction uses a nonstandard relative positional encoding that assumes the space bound is known in advance; how can positional encodings be designed to adapt dynamically when the space bound is unknown? (iii) Our results focuses only the expressiveness. It remains an open question whether standard training protocols can actually discover such behavior; and empirical experiments are still to be done.

---

[6]Here, for convenience, we allow the front head to point to negative indices; whenever this occurs, the corresponding cell is assumed to contain the blank symbol $\perp$.

## ACKNOWLEDGMENTS

Qian Li's work was supported by Hetao Shenzhen-Hong Kong Science and Technology Innovation Cooperation Zone Project (No.HZQSWS-KCCYB-2024016), and Guangdong Basic and Applied Basic Research Foundation (Grant No.2026A1515011078). Yuyi Wang's work was supported by the Hunan Provincial Natural Science Foundation (Grant No.2024JJ5128).

**Reproducibility Statement.** This paper is theoretical. We make all assumptions, model definitions, and notation explicit in Section 2 and Appendix A, present a complete statement of our main results in Theorem 1, and provide full proofs of every lemma and theorem in Sections 3, 4, and Appendix B.

**LLM usage.** LLMs is used to aid and polish writing.

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

## A Modifications on Transformers in Turing Completeness Proofs

Existing proofs of Turing-completeness for Transformers typically do not employ the vanilla architecture, but instead introduce specific modifications. For clarity, we summarize these changes below.

Pérez et al. (2021) consider a Transformer with an absolute positional encoding given by the triplet $(i, 1/i, 1/i^2)$ at position $i$. In their construction, the attention score is defined as the negative absolute value of the dot product, and the attention mechanism uses average-hard attention. Moreover, the feed-forward layers employ sigmoid activations in place of ReLUs.

Merrill & Sabharwal (2024) dispense with positional encodings altogether and instead adopt *Strict Causal Masking*, where attention at position $i$ can access all tokens up to position $i-1$ but not the current token $i$. Their construction also uses average-hard attention and introduces a *Projected Pre-Norm*: an arbitrary linear projection is allowed before layer normalization.

In the proof of Li et al. (2024), the simulated circuit is directly encoded into the absolute positional encoding, rather than into the parameters.

Qiu et al. (2024) consider Transformers with nonstandard absolute positional encodings. In their construction, the query, key, and value maps in the attention sublayer are implemented as ReLU networks rather than linear transformations.

Finally, both Li & Wang (2025) and this work employ nonstandard *relative* positional encodings.

## B Proof of Theorem 3

*Proof.* Let $\mathsf{TM} = (\Sigma, Q, \Delta)$ be a synchronous $K$-queue Turing machine running in $t(n)$ time and $\leq s(n)$ space. Without loss of generality, let the tape alphabet be $\Sigma = \{0, 1, \perp\}$ and the set of states be $Q = \{0,1\}^c$ for some $c \in \mathbb{N}$. We further assume that $q_{start}$ never appears in any transition $\delta(\sigma_0, \cdots, \sigma_{K-1}, q)$, i.e., once $\mathsf{TM}$ leaves $q_{start}$ at the beginning, it never returns. Because each of the $K$ queues pops exactly one element and also appends exactly one element at each step, their sizes are fixed during the execution of $\mathsf{TM}$. Let $s_0(n), s_1(n), \cdots, s_{K-1}(n)$ denote the queue sizes, with $s_{\max}(n) := \max_{r=0}^{K-1} s_r(n)$. For technical convenience, without loss of generality, we assume $s_1(n), s_2(n), \cdots, s_K(n)$ are always even numbers.

We now construct a constant bit-size Transformer $\mathsf{TF}$ with a context window of length $s(n)$ to faithfully simulate $\mathsf{TM}$ step by step. Let $\mathcal{V} = \Sigma^K \times Q \times P = \{0, 1, \perp\}^K \times \{0,1\}^c \times \{0,1\}$, where $P$ is a bit indicating the parity of the location. The intuition is as follows:

**1. Token embedding layer** Map each token $v = (\boldsymbol{\sigma} := (\sigma_0, \cdots, \sigma_{K-1}), q, p)$ to a vector $\mathsf{emb}(\boldsymbol{\sigma}, q, p) := (\mathsf{emb}_0(\sigma_0, q, p), \cdots, \mathsf{emb}_{K-1}(\sigma_{K-1}, q, p)) \in \mathbb{R}^{(c+9) \times K}$ as follows:

$$\mathsf{emb}_r(\sigma_r, q) = \begin{cases} (1, 1, 0, 0, q, 0, 0, 0, p, 0), & \text{if } \sigma_r = 0; \\ (1, 0, 1, 0, q, 0, 0, 0, p, 0), & \text{if } \sigma_r = 1; \\ (1, 0, 0, 1, q, 0, 0, 0, p, 0), & \text{if } \sigma_r = \perp. \end{cases} \tag{1}$$

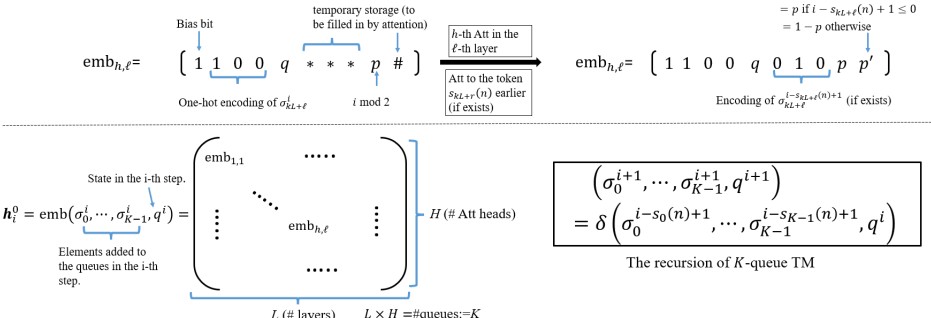

Figure 2: Illustration of our construction in the proof of Theorem 3.

**2. Relative positional encoding**   Inside the attention score, add relative information as follows[7]:

$$\mathsf{pos}(\Delta) \in \mathbb{R}^{d/H} = \begin{cases} \mathbf{1}_{d/H}, & \text{if } \Delta = 0, \\ 2\sum_{r:\Delta=s_r(n)-1} \boldsymbol{e}_r, & \text{if } \Delta > 0. \end{cases} \tag{2}$$

where $\boldsymbol{e}_r \in \mathbb{R}^{d/H}$ stands for the standard basis vector with a 1 at the $r$-th coordinate.

**3. Decoder layer**   The Transformer has $L$ decoder layers, and each decoder layer has $H$ attention heads, with $H \cdot L = K$. For the $k$-th attention head in the $\ell$-th layer, where $k = 0, \cdots, H-1$ and $\ell = 0, \cdots, L-1$, we set the matrix $\boldsymbol{K}_k^\ell, \boldsymbol{Q}_k^\ell, \boldsymbol{V}_k^\ell \in \mathbb{R}^{d \times d/H} = \mathbb{R}^{d \times ((c+9) \times L)}$ where $d = (c+9) \cdot K$ as follows: let $r = k \cdot L + \ell$,

- Matrix $\boldsymbol{K}_k^\ell$: all-zeros matrix;

- Matrix $\boldsymbol{Q}_k^\ell$: it has only one non-zero entry $Q_{(c+9)r+1,r} = 1$;

- Matrix $\boldsymbol{V}_k^\ell$: it has four non-zero entries: $V_{(c+9)r+2,(c+9)\ell+c+5} = V_{(c+9)r+3,(c+9)\ell+c+6} = V_{(c+9)r+4,(c+9)\ell+c+7} = V_{(c+9)r+c+8,(c+9)(\ell+1)} = 1$.

We set $\boldsymbol{W}^\ell \in \mathbb{R}^{d \times d}$ as the identity matrix and set the bias vector $\boldsymbol{b}$ to $\mathbf{0}_d$.

The feed-forward network $\mathsf{FF}^\ell$ maps $\boldsymbol{h}^\ell$ to $\mathbf{0}_d$, except that $\mathsf{FF}^{L-1}$ is designed to simulate the transition function $\delta$. Specifically, we let

$$\mathsf{FF}^{L-1}(\boldsymbol{h}) + \boldsymbol{h} = \boldsymbol{e}_{(\delta(\sigma_1,\cdots,\sigma_K,q)),p'}, \text{ where } q = \boldsymbol{h}_{5:c+4} \text{ and } p' = 1 - \boldsymbol{h}_{c+8} \tag{3}$$

$$\sigma_r = \begin{cases} \perp, & \text{if } \boldsymbol{h}_{(c+9)r+c+8} = \boldsymbol{h}_{(c+9)r+c+9}; \\ h_{(c+9)r+c+5:(c+9)r+c+7}, & \text{otherwise}; \end{cases}$$

Here, $\boldsymbol{e}_{(\boldsymbol{\sigma},q,p)} \in \mathbb{R}^{|\mathcal{V}|}$ denotes the standard basis vector with a 1 at the coordinate indexed by $(\boldsymbol{\sigma}, q, p)$.

**4. Output layer**   We set $W^{\mathsf{out}}$ to be the identity matrix and $\boldsymbol{b}^{\mathsf{out}}$ the all-zeros vector. Then the output layer outputs $v_{i+1} := \arg\max(\boldsymbol{h}_i^L)$.

**Analysis**   What remains is to prove that our construction TF faithfully simulates TM. Let $(\boldsymbol{\sigma}^1, q^1), (\boldsymbol{\sigma}^2, q^2), \cdots$ denote the execution log of TM running on $x \in \{0,1\}^n$, where $\boldsymbol{\sigma}^i = (\sigma_0^i, \cdots, \sigma_{K-1}^i)$ denote the elements added to the queues in the $i$-th step and $q^i$ is the state when adding $\boldsymbol{\sigma}^i$. For convenience, we assume initially the input queue ($r = 0$) contains $x$, and the other $K-1$ queues contain only blank symbols $\perp$; that is, for $i \leq n$, $(\boldsymbol{\sigma}^i, q^i) = (x_i, \perp^{K-1}, q_{start})$. Then, one can see that

$$(\boldsymbol{\sigma}^{i+1}, q^{i+1}) = \delta\left(\sigma_0^{i-s_0(n)+1}, \cdots, \sigma_{K-1}^{i-s_{K-1}(n)+1}, q^i\right),$$

---

[7]Without loss of generality, we assume $d/H \geq H \cdot L$ or equivalently $c + 9 \geq H$ by making $c$ sufficiently large.

where if $i - s_r(n) + 1 \leq 0$, then we set

$$\sigma_r^{i-s_r(n)+1} = \begin{cases} x_i, & \text{if } r = 0; \\ \perp, & \text{if } r = 1, 2, \cdots, K-1. \end{cases}$$

Let $v_1, v_2, \ldots$ denote the token sequence in the context of TF when it takes $v_1 = (x_1, \perp^{K-1}, q_{start}, 1), \cdots, v_n = (x_n, \perp^{K-1}, q_{start}, n \mod 2)$ as input. By induction, we will show that for each $i \geq 1$, we have $v_i = (\boldsymbol{\sigma}^i, q^i, i \mod 2)$, and then finish the proof. The base case $i \leq n$ is trivial. Now, we assume $i \geq n+1$ and $v_{i'} = (\boldsymbol{\sigma}^{i'}, q^{i'}, i' \mod 2)$ for any $i' \leq i$.

**Claim 2.** *For any $0 \leq \ell \leq L-1$, $\boldsymbol{h}_i^\ell$ and $\boldsymbol{h}_i^0$ differ only in the entries at indices $(c+9) \cdot r + c + 5 : (c+9) \cdot r + c + 7$ and $(c+9) \cdot r + c + 9$.*

*Proof.* By the description of TF, we have

$$h_i^\ell = \mathsf{FF}(h^{\ell-0.5}) + h_i^{\ell-0.5} = h_i^{\ell-0.5} = h_i^{\ell-1} + \left( (\boldsymbol{a}_{i,1}^{\ell-1})^T, \cdots, (\boldsymbol{a}_{i,H}^{\ell-1})^T \right)^T,$$

where

$$\boldsymbol{a}_{i,k}^{\ell-1} = s_{k,i,1}^{\ell-1} \cdot \boldsymbol{h}_1^{\ell-1} \cdot \boldsymbol{V}_k^{\ell-1} + \cdots + s_{k,i,i}^{\ell-1} \cdot \boldsymbol{h}_i^{\ell-1} \cdot \boldsymbol{V}_k^{\ell-1}$$

By definition of $\boldsymbol{V}_k^{\ell-1}$, $\boldsymbol{a}_{i,k}^{\ell-1}$ is zero at all indices except $(c+9) \cdot r + c + 5 : (c+9) \cdot r + c + 7$ and $(c+9) \cdot r + c + 9$. By induction on $\ell$, we get the claim. $\square$

**Claim 3.** $\boldsymbol{h}_{(c+9)r+c+8} = \boldsymbol{h}_{(c+9)r+c+9}$ *if and only if* $i - s_r(n) + 1 \leq 0$. *Furthermore, if* $i - s_r(n) + 1 > 0$, *then* $\boldsymbol{h}_{i,(c+9)r+c+5:(c+9)r+c+7}^{L-0.5} = \sigma_{i-s_r(n)+1}$.

*Proof.* Let $(\ell, k)$ be the unique pair satisfying that $r = k \cdot L + \ell$. One can easily check that

$$\boldsymbol{h}_{i,(c+9)r+1:(c+9)(r+1)}^{L-0.5} = \boldsymbol{h}_{i,(c+9)r+1:(c+9)(r+1)}^0 + \left( \boldsymbol{a}_{i,k,(c+9)\ell+1:(c+9)(\ell+1)}^\ell \right)^T.$$

For the $k$-th attention head in the $\ell$-th layer, by Claim 2, we have $\boldsymbol{K}_k^\ell \cdot \boldsymbol{h}_j^\ell = \boldsymbol{0}_{d/H}$, $\boldsymbol{Q}_k^\ell \cdot \boldsymbol{h}_i^\ell = \boldsymbol{Q}_k^\ell \cdot \boldsymbol{h}_i^0 = \boldsymbol{e}_r$, so

$$\boldsymbol{s}_{k,i}^\ell = \mathsf{hardmax} \left( \langle \mathsf{pos}(i-1), \boldsymbol{h}_i^0 \cdot \boldsymbol{Q}_k^\ell \rangle, \cdots, \langle \mathsf{pos}(i-i), \boldsymbol{h}_i^0 \cdot \boldsymbol{Q}_k^\ell \rangle \right)$$

$$= \begin{cases} (0, \cdots, 0, 1) \in \mathbb{R}^i, & \text{if } i \leq s_r(n) - 1, \\ (\boldsymbol{0}_{i-s_r(n)}, 1, \boldsymbol{0}_{s_r(n)-1}) \in \mathbb{R}^i, & \text{if } s_r(n) \leq i \leq s(n) - 1, \\ (\boldsymbol{0}_{s(n)-s_r(n)}, 1, \boldsymbol{0}_{s_r(n)-1}) \in \mathbb{R}^{s(n)}, & \text{if } i \geq s(n). \end{cases}$$

and

$$\boldsymbol{a}_{k,i}^\ell = \sum_{j=1}^i s_{k,i,j}^\ell \cdot \left( \boldsymbol{h}_j^\ell \cdot \boldsymbol{V}_k^\ell \right) = \sum_{j=1}^i s_{k,i,j}^\ell \cdot \left( \boldsymbol{h}_j^0 \cdot \boldsymbol{V}_k^\ell \right)$$

$$= \sum_{j=1}^i s_{k,i,j}^\ell \cdot \left( \boldsymbol{0}_{(c+9)\ell+c+4}, \boldsymbol{h}_{j,(c+9)\ell+2:(c+9)\ell+4}^0, 0, \boldsymbol{h}_{j,(c+9)\ell+c+8}^0, \boldsymbol{0}_{(c+9)(L-\ell-1)} \right)$$

$$= \begin{cases} \left( \boldsymbol{0}_{(c+9)\ell+c+4}, \boldsymbol{h}_{i,(c+9)\ell+2:(c+9)\ell+4}^0, 0, \boldsymbol{h}_{i,(c+9)\ell+c+8}^0, \boldsymbol{0}_{(c+9)(L-\ell-1)} \right), & \text{if } i \leq s_r(n) - 1, \\ \left( \boldsymbol{0}_{(c+9)\ell+c+4}, \boldsymbol{h}_{i-s_r(n)+1,(c+9)\ell+2:(c+9)\ell+4}^0, 0, \boldsymbol{h}_{i-s_r(n)+1,(c+9)\ell+c+8}^0, \boldsymbol{0}_{(c+9)(L-\ell-1)} \right), & \text{otherwise} \end{cases}$$

Recall the assumption $s_1(n), s_2(n), \cdots, s_K(n)$ are always even numbers, now the claim is clear by the induction hypothesis. $\square$

Besides, in the last feed-forward network layer $\mathsf{FF}^{L-1}$, we have

$$\boldsymbol{h}_{i,5:c+4}^{L-0.5} = \boldsymbol{h}_{i,5:c+4}^0 = q_i, \text{ and } \boldsymbol{h}_{i,(c+9)r+c+5:(c+9)r+c+7}^{L-0.5} = \sigma_{i-s_r(n)+1}.$$

The vector $\boldsymbol{h}_i^{L-0.5}$ is mapped to $\mathsf{FF}(\boldsymbol{h}_i^{L-0.5}) + \boldsymbol{h}_i^{L-0.5}$, which is $\boldsymbol{e}_{\delta\left(\sigma_0^{i-s_0(n)+1}, \cdots, \sigma_{K-1}^{i-s_{K-1}(n)+1}, q^i\right)}$ according to Equation (3). Then one can see that the output layer outputs $(\boldsymbol{\sigma}^{i+1}, q^{i+1})$ as desired. $\square$

