# OpenReview forum: "Efficient Turing Machine Simulation with Transformers"
_ICLR.cc/2026/Conference — ICLR 2026 Poster_

### Official Review · Reviewer_vfEe · 2025-10-19

**Soundness:** 4
**Presentation:** 4
**Contribution:** 4
**Rating:** 10
**Confidence:** 4

**Summary:**

This paper presents a theoretical advance in understanding the computational efficiency of Transformers.
It proves that any $(t(n), s(n))$-bounded multi-tape Turing Machine can be simulated by a constant bit-size Transformer with an optimal $O(s(n))$ context window and only $O(s(n)^c)$ chain-of-thought (CoT) steps per simulated step, where $c > 0$ can be made arbitrarily small.
The work also introduces geometric-offset sparse attention, linking theoretical universality with practical efficiency.
The key technical innovation lies in efficiently simulating multi-tape TMs via synchronous multi-queue TMs, followed by a Transformer-based realization.

**Strengths:**

Improves efficiency bounds for Turing-complete Transformer simulations from $O(s(n))$ to $O(s(n)^c)$ for any small $c > 0$.

The synchronous multi-queue TM construction is original and cleverly designed.

The geometric-offset attention model aligns with sparse attention mechanisms used in practice (e.g., LogSparse, PowerAttention).

**Weaknesses:**

A small experimental validation or simulation would make the theoretical results more tangible.

Several sections (especially Step I and Appendix B) are heavy in notation and could benefit from clearer examples or figures.

**Questions:**

Could the per-step overhead be further reduced to $O(1)$, and what are the theoretical obstacles?

How might geometric-offset attention be implemented or approximated in modern Transformers with dynamic context lengths?

Could a simplified Transformer experiment empirically demonstrate the predicted efficiency?

---

> ### Author Response · Authors · 2025-11-21
>
> We greatly appreciate the reviewer’s recommendation and constructive comments!
>
> **Q1:** A small experimental validation or simulation would make the theoretical results more tangible. Could a simplified Transformer experiment empirically demonstrate the predicted efficiency?
>
> **A1:** We agree that empirical experiments would be valuable and make the results more tangible. We highlight it as a future direction in the revision.
>
>
>
> **Q2:** Several sections (especially Step I and Appendix B) are heavy in notation and could benefit from clearer examples or figures.
>
> **A2:** In the revision, we have added **Figure 1** to illustrate Step I and **Figure 2** to illustrate Step II, hoping they make the related parts more clearer.
>
>
>
> **Q3:** Could the per-step overhead be further reduced to $O(1)$, and what are the theoretical obstacles?
>
> **A3:** We do not know how to achieve $O(1)$ per-step overhead under the same constraints (constant bit-size and an $O(s(n))$ window); this remains an open problem. With our current multi-queue framework, Huber (1993) showed that online simulation of 2-tape TM on $k$-queue TM requires time $\tilde{\Omega}(t^{1+1/k})$, which suggests that achieving $O(1)$ overhead would likely require a different intermediate computational model.
>
> On the other hand, we do not have a lower bound ruling out $O(1)$ overhead, and proving such a lower bound would likely require new techniques.
>
>
>
> **Q4:** How might geometric-offset attention be implemented or approximated in modern Transformers with dynamic context lengths?
>
> **A4:** We sincerely thank the reviewer for this constructive comment. In Theorem 1, the geometric offsets used by our transformer construction are
> $$
> \lceil s(n)^{1/k'}\rceil, \lceil s(n)^{1/k'}\rceil^2,\dots,\lceil s(n)^{1/k'}\rceil^{k'},
> $$
> with $k' = K/(6k)$. Thus the common ratio is $\lceil s(n)^{1/k'}\rceil=\lceil s(n)^{6k/K}\rceil$.
>
> In the regime of practical interest, the head-layer product $K$ is typically large (e.g. $K$ on the order of $10^3-10^4$ in modern LLMs). Moreover, it is natural to focus on $k=2$, since $2$-tape TM can efficiently simulate multi-tape TMs with only a logarithmic slowdown. So, if we take $K=6\times 10^3$ and $k=2$, then the **common ratio would be $\lceil s(n)^{6k/K}\rceil=2$ for any $s(n)\leq 2^{500}$.**
>
>  Motivated by these observations, for practical use, we propose to **simply fix the relative offset to $\{1,2,4,8,\cdots\}$**, truncated according to the available dynamic window size $W$.
>
> In the revision, we included the above discussion in Page 2.

---

### Official Review · Reviewer_Carr · 2025-10-30

**Soundness:** 3
**Presentation:** 2
**Contribution:** 2
**Rating:** 4
**Confidence:** 5

**Summary:**

This is a theoretical work whose main contribution is the proof that any (t(n), s(n)) time-space bounded many-tape Turing Machine can be simulated by a constant bit-size Transformer with context window of optimal length and using O(s(n)^c), with any c>0, chain-of-thought (CoT) steps for a singe simulation step. This improves on previous results regarding efficient TM simulations by Transformers. To show this result, the authors first present a time-efficient simulation of k-tape Turing Machines using so-called “synchronous” queue TMs and then describe a simulation of multi-queue TMs by Transformers. The proposed simulation of TMs using synchronous queues seems to be the main technical achievement of this paper.

**Strengths:**

This work improves on previously presented results on efficient TM simulations by constant bit-size Transformers. In particular, it is shown how to reduce the number of CoT steps per simulated step from O(s(n)) to O(s(n)^c), for any constant c>0. The analysis is non-trivial and the result is in line with what is actively being researched by many authors studying Transformers through the lens of computational complexity.

**Weaknesses:**

Although the paper improves on known results, the work presents incremental progress in the field. In particular, as shown in Table 1, the submitted paper improves on the results of Li and Wang (2025) essentially only with respect to the number of CoT per single TM step. Importantly, the simulation results (presented in Table 1) do not take into account the total CoT length of the simulation. So, by simulating a (t(n), s(n)) time-space bounded multi-tape TMs with a single-tape that increases time but leaves space unchanged, we can conclude that the result given in the line “Li & Wang (2025)” holds true for multi-tape TM.

The main technical achievement of this paper seems to be the simulation of TMs by TMs using synchronous queues. But the idea behind this construction is not entirely new: as the authors note, the proof relies on the general idea of ​​Theorem 3.2 in (Hühne, 1993). The modification proposed in the submitted paper is somewhat non-obvious, but in my opinion it is not strong enough for ICLR.

It would be nice to have *precise* descriptions for the key notions used in the paper as, e.g., precision, CoT, Window, embedding dimension (Dim.).

**Questions:**

According to the proof, it seems that in the statement of Theorem 1 the total length of the CoT of the simulation should be $O(t(n) \cdot s(n)^{1/k'} \cdot s(n)^{6k/K})$, not $O(t(n) \cdot s(n)^{6k/K})$, where $k'$ is used in Theorem 2.

According to your description L. 182-186 a TM cannot empty the stack.

L. 310: what do you mean by "rotates the tape heads"?

Use uniformly throughout the document: stack or queue.

---

> ### Author Response · Authors · 2025-11-21
>
> We appreciate the valuable comments and feedback. We provide a detailed clarification hoping to address the reviewer’s concerns.
>
> **Q1:** Although the paper improves on known results, the work presents incremental progress in the field.
>
> **A1:** We agree that this work builds on Li & Wang (2025), so in that sense it is incremental. At the same time, we would like to clarify our main contributions and novelties:
>
> - Under the same regime (constant bit-size and an $O(s(n))$ window length), our main theorem reduces the CoT overhead per TM step from $s(n)$ to $s(n)^c$, consequently, reduces the total CoT length from $t(n)s(n)$ to $t(n)s(n)^c$.
> - Our transformer construction also suggests geometric-offset attention as a promising architectural direction. **In particular, in the regime of practical interest, where $s(n)\leq 2^{500}$, the geometric offsets used by our transformer construction in Theorem 1 reduce to $\{1,2,4,8,\cdots\}$, truncated according to the window size.**
> - Our main technical novelty lies in Step 1 (from multi-tape TM to synchronous multi-queue TM). We acknowledge that Step 2 (from synchronous multi-queue TM to Transformers) is a incremental improvement over Li & Wang (2025).
>
>
>
> **Q2:** As shown in Table 1, the submitted paper improves on the results of Li and Wang (2025) essentially only with respect to the number of CoT per single TM step. Importantly, the simulation results (presented in Table 1) do not take into account the total CoT length of the simulation.  So, by simulating a (t(n), s(n)) time-space bounded multi-tape TMs with a single-tape that increases time but leaves space unchanged, we can conclude that the result given in the line “Li & Wang (2025)” holds true for multi-tape TM.
>
> **A2:** We thank the reviewer for pointing this out. We agree that it is more appropriate to compare the **total** CoT length rather than the CoT overhead per TM step, and we have revised Table 1 accordingly.
>
> Specifically, let us focus on the efficiency of Transformers for simulating a **$t$-time, $s$-space multi-tape TM**:
>
> - **Li & Wang (2025):** By a standard technique, such a multi-tape TM can be converted into a Post machine running in time $O(t⋅s)$ and space $O(s)$. Their proof then shows that this Post machine can be simulated by a Transformer with total CoT length $O(t⋅s)$ and window size $O(s)$.
> - **This work:** Our Transformer construction simulates the same multi-tape TM using total CoT length $O(t⋅s^c)$ and window size $O(s)$.
>
>
>
> **Q3:** The main technical achievement of this paper seems to be the simulation of TMs by TMs using synchronous queues. But the idea behind this construction is not entirely new: as the authors note, the proof relies on the general idea of Theorem 3.2 in (Hühne, 1993). The modification proposed in the submitted paper is somewhat non-obvious, but in my opinion it is not strong enough for ICLR.
>
> **A3:**  We fully agree that our Step 1 exploits the idea from (Huhne, 1993) of using queues of geometrically increasing size, and we explicitly acknowledge this in Remark 1 in the original version.
>
> However, Huhne’s result is proved for a **more permissive queue model** in which each queue may remain idle in a step (i.e., neither pop nor push), while our setting requires a **strictly synchronous** machine where every queue must pop and append exactly one symbol at every step. This synchrony requirement significantly complicates the simulation: when transferring data between adjacent levels, the two participating heads must reach the relevant cells **simultaneously**.
>
> **The main technical work in Step 1 is to redesign the simulation so that it satisfies these synchrony constraints while preserving the desired time and space bounds.** This includes, for example, introducing auxiliary buffer queues, splitting a content queue into two half-queues, and maintaining new balance invariants that ensure correct head alignment during PUSH/POP operations.
>
>
>
> **Q4:** It would be nice to have *precise* descriptions for the key notions used in the paper as, e.g., precision, CoT, Window, embedding dimension (Dim.).
>
> **A4:** Added as suggested. Concretely,
>
> - **Precision** $p$: all learnable parameters are stored with $p$ bits per scalar; the **bit-size** of a Transformer is then $p\times|\theta|$, where $|\theta|$ is the number of all learnable parameters.
> - **CoT length:** the total number of tokens generated.
> - **Window:** A Transformer is said to have a context window of length $s$ if the query can attend only to the most recent $s$ tokens.
> - **Embedding dimension:** the dimension of hidden embedding vectors $h_i^\ell$.

---

> ### Author Response · Authors · 2025-11-21
>
> **Q5:** According to the proof, it seems that in the statement of Theorem 1 the total length of the CoT of the simulation should be $O(t(n)\cdot s(n)^{1/k'}\cdot s(n)^{6k/K})$, not $O(t(n)\cdot s(n)^{6k/K})$, where is used in Theorem 2.
>
> **A5:** It should be $O(t(n)\cdot s(n)^{6k/K})$.  Specifically, our proof has two steps:
>
> - Step 1 (Theorem 2): Any $t$-time $s$-space $k$-tape TM  can be simulated by a synchronous $6kk'$-queue TM running in $O(t\cdot s^{1/k'})$-time and $O(s)$-space.
> - Step 2 (Theorem 3):  A synchronous $K$-queue TM running in $t'$-time and $s$-space can be simulated by a constant bit-size Transformer with head–layer product $K$, context window $O(s)$, and CoT length $O(t')$.
>
> In Theorem 1, we aim to simulate $k$-tape TM with a Transformer whose head-layer product is $K$. We therefore set $K=6kk'$, i.e., $1/k'=6k/K$. Combining these two steps, the total CoT length is $O(t')=O(t\cdot s^{1/k'})=O(t\cdot s^{6k/K})$.
>
>
>
> **Q6:** Use uniformly throughout the document: stack or queue.
>
> According to your description L. 182-186 a TM cannot empty the stack.
>
> **A6:** On **Stack vs. queue**: In Step 1, we simulate **each TM tape using two stacks**, and then **simulate each stack with a collection of queues**.  In other words, stacks serve as a bridge between tape and queue. **In this paper, "queue" and "stack" refer to different things.**
>
> On **L182-186**: The phrase "the other $k−1$ queues contains only blank symbols" does **not** mean that these queues are physically empty. A queue is considered **logically empty** when all its cells store distinguished blank symbol, and we never meet a length-zero queue.
>
>
>
> **Q7**:  L. 310: what do you mean by "rotates the tape heads"?
>
> **A7:**  In Step 1, we treat each queue as a special kind of tape of the same length, where the tape head moves exactly one cell to the right in a cyclic fashion at every step (see the left part of Figure 1 for an illustration). Thus, “rotates the tape heads” means that "shifts the tape head to the right cyclically" (or equivalently, rotates the queue).

---

> > ### Comment · Reviewer_Carr · 2025-11-25
> > **Further questions and concerns**
> >
> > Dear Authors, thank you very much for your responses and comments, which largely address my (and other reviewers') questions. Below, I would like to present my further concerns regarding your paper.
> >
> > The important complexity measure used in your results is the space complexity of the (multi-tape) Turing machines (TMs), which affects the results of the transformers’ complexity. In particular, in Theorem 1 you get window length $O(s(n))$ and the number of CoT steps for each simulated TM step as $O(s(n)^{6k/K})$, etc. Unfortunately, in your paper you do not explicitly define the space complexity of TMs. According to the standard definition used in computational complexity theory, we define the space complexity of a k-tape TM (which has one read-only input and k-1 working tapes, as assumed in your paper) as the number of different cells visited on k-1 work tapes during the computation. This allows to consider sublinear space complexities $s(n)$ such as, e.g., $\log n, \log^2 n, \sqrt{n}$. On the other hand, from the proof of Theorem 2 (which is crucial) it follows, that a $(t(n), s(n))$ time-space bounded k-tape TM M can be simulated by a synchronous $6kk'$-queue TM M' which is $(O(t(n) \cdot S(n)^{1/k'}),O(S(n)))$ time-space bounded, where $S(n):=\max\{n,s(n)\}$, and not as stated in the paper that M' is $(O(t(n) \cdot s(n)^{1/k'}),O(s(n)))$ time-space bounded. This has an impact on many of the results mentioned in the paper. In particular in Theorem 1 the correct window length is $O(\max\{n,s(n)\})$ and not $O(s(n))$, and the number of CoT steps for each simulated TM step is $O(\max\{n,s(n)^{6k/K}\})$, and not $O(s(n)^{6k/K})$, etc.
> >
> > As you explain in your rebuttal,  the main technical novelty of the paper lies in Step 1 (from multi-tape TM to synchronous multi-queue TM). However, it seems that you are significantly worsening the space complexity in this simulation: from $s(n)$  to $\max\{n,s(n)\}$. Could you please address these concerns?
> >
> > Regarding my comment in the first round: "provide precise descriptions for the key notions". I meant that these definitions should be given in the paper so that the reader has clarity. Also, the explanation of the use of stack versus queue should be given directly in the paper.

---

> ### Author Response · Authors · 2025-11-26
>
> Thanks for the constructive and valuable comments.
>
>
>
> **Q1:** The important complexity measure used......and not $O(s(n)^{6k/K})$, etc.
>
> **A1:** Yes, you are right. If we define the space complexity as the total number of cells used by the $k-1$ working tapes, we should use $\max\{s(n),n\}$ instead of $s(n)$ in the theorems.
>
>
>
> In the original version, we intended to uses the following multi-tape TM definition:
>
> - Input tape: both readable and writable.
>
> - Space complexity: counts cells on all tapes, including the input tape.
>
> With this definition, the theorems remains unchanged.
>
>
>
> Thanks again for pointing out this important issue. We have corrected the related part accordingly.
>
>
>
> **Q2:** As you explain in your rebuttal, the main technical novelty of the paper lies in Step 1 (from multi-tape TM to synchronous multi-queue TM)... Could you please address these concerns?
>
> **A2:** If we assume that the input tape/queue is read-only, and define the space complexity as the total number of cells used by the working tapes/queues, then Theorem 2 still essentially holds even when $s(n)=o(n)$ if we additionally assume that: at the start of the computation, the input tape head moves one cell to the right at each step until the entire input has been read; thereafter it stays for the rest of the computation (intuitively, the input tape behaves like a queue).
>
>
>
> In addition, we want to remark that: nevertheless, the window must be at least as long as the input length $n$, since otherwise the Transformer cannot even access the entire input.
>
> More specifically, note that the entire input string is presented before the Transformer starts generating CoT tokens, and at every step the transformer can look back only $W$ positions. If $W<n$, the first $n-W$ input symbols are permanently outside the transformer's attention and can never influence any CoT token.
>
>
>
> **Q3:** Regarding my comment in the first round: "provide precise descriptions for the key notions". I meant that these definitions should be given in the paper so that the reader has clarity. Also, the explanation of the use of stack versus queue should be given directly in the paper.
>
> **A3:** We have revised the relevant parts accordingly.

---

### Official Review · Reviewer_VMLy · 2025-10-30

**Soundness:** 4
**Presentation:** 4
**Contribution:** 3
**Rating:** 8
**Confidence:** 4

**Summary:**

Summary: this paper presents a refined theoretical analysis of the ability of CoT transformers to simulate Turing machines, which has been studied in prior work. The paper shows that a multitape TM with time t(n) and space s(n) can be simulated by a constant bit-size transformer with context length O(s) and runtime O(t * s^c), for arbitrarily small c. Thus, constant bit-size transformers can quite effectively simulate transformers. The proof technique is interesting in its own right, going through a conversion of multitape Turing machines to multiqueue Turing machines.

**Strengths:**

1. I find the main result about the space requirements needed to simulate a Turing machine with constant-bit-size CoT transformers to be valuable.
2. The intermediate result converting multitape Turing machines to multiqueue Turing machines is interesting in its own right and technically innovative.
3. The high-level technical plan and proofs are clear and rigorous

**Weaknesses:**

### Make Dependence of Space/Context Window on k' Explicit

In theorem 2, how does the space O(s) depend on the queue factor k'? The way you are reducing time overhead is increasing k', so it would be nice to understand how space scales with this.

It would also be good to understand how this shows up in the main result about transformers: you say that we can make the time overhead arbitrarily small, but how does this increase the context window we need?

### Theorem 3 Suggestions

Overall, the theorem looks solid, but I have some minor suggestions for improvement:
- Clarify non-standard positional encodings in theorem statement
- Clarify use of hardmax in theorem statement; also it would be helpful to explicitly reference unique hard (UHAT) vs. averaging hard (AHAT) variants of hardmax. presumably your construction should work with either
- How uniform is this construction? (see below)

I understand constant bit-size (a la Li and Wang) to mean that the number of params and precision independent of context length. Does this imply that the transformer constructed in Theorem 3 is fully uniform, i.e., the parameters can't change at all with n? It would be helpful to explicitly mention the level of parameter uniformity that your construction attains (potentially in Table 1). E.g., the Merrill & Sabharwal's construction re-uses the same parameters for any n, so it is fully uniform, but this is not true for all constructions (e.g., Li et al., where position embeddings can evolve in a complicated way with n).

### Quadratic vs. Linear-Time Attention Discussion Deserves More Nuance

> This shows that the common argument “quadratic-time attention= ⇒fundamental throughput bottleneck” against Transformer-based AGI may not be a principled limitation but a byproduct of dense attention.

This sets up a bit of a strawman and makes a vague/strong claim against it. Can you clarify the argument in scare quotes (and attribute to someone) and clarify your counterargument. E.g., you're saying that expressive power does not require quadratic-time attention?

Moreover, while quadratic time attention is sometimes invoked as a drawback of attention, the related issue that matters more in practice (e.g., for GPU memory limitations) is the linear memory incurred by attention on long sequences to store the KV cache. This limitation is not overcome by your construction (and, in fact, is more fundamental), since for any function requiring s(n) = Omega(n), you will need to store linear memory to compute it.

**Questions:**

See "Weaknesses"

---

> ### Author Response · Authors · 2025-11-21
>
> We sincerely thank the reviewer for the valuable comments and feedback.
>
> **Q1:** Make Dependence of Space/Context Window on $k'$ Explicit.
>
> **A1:** In Theorem 2, the total space used by the multi-queue machine is
> $$
> 2k\cdot \sum_{i=1}^{k'} 4\times \lceil s(n)\rceil^{i/k'}\leq 8k\cdot \sum_{i=1}^{k'} (s(n)+1)^{i/k'}\leq 8k\cdot 2(s(n)+1)=O(ks(n)).
> $$
> In Theorem 3, the context window length is chosen to be the maximum queue length, which equals $\lceil s(n)\rceil^{k'/k'}\leq s(n)+1$. Hence, in Theorem 1, the required context window length is at most $s(n)+1$.
>
> In the revision, we updated the relevant parts to make this dependence explicit.
>
>
>
> **Q2:** Theorem 3 Suggestions.
>
> **A2:** Thank you for these suggestions; we have updated the related parts accordingly.
>
> - **Relative PE.** We use a unlearnable relative PE, defined explicitly in Equation (2).
>
> - **The choice of hardmax.**  In our Transformer construction, the input to hardmax is always a one-hot vector, and so is the hardmax output. Hence, our construction works with both UHAT and AHAT.
>
> - **Uniformity.**
>
>   - Following Li & Wang (2025), our relative PE is unlearnable, and evolves with $n$ as defined in Equation 2.
>
>   - To simulate the given TM on longer inputs, we only need to **adjust the relative PE** according to Equation (2); **all learnable parameters (embedding, QKV matrices, FFN, etc.), including their precision, remains unchanged.**  Thus, for a given TM, there is a single learnable parameter vector that is reused for all input lengths.
>
>   - For further discussion of this point and our notion of uniformity/constant bit-size, we also refer the reviewer to our answer A2 to Reviewer iCy5.
>
>
>
> **Q3:** Quadratic vs. Linear-Time Attention Discussion Deserves More Nuance
>
> **A3:** Rather than to introduce a new formal argument, our intention was to summarize the very broad concern about the quadratic **time complexity** of full attention, which poses a critical bottleneck for scaling transformers to long contexts.
>
> **On memory costs.** We fully agree with the reviewer that, the memory issue matters more in practice, and our construction does not overcome the memory limitation, as it stores the whole $s(n)$-long window.
>
> In the revision, we instead write that our results "suggests that the wide concern that the quadratic time complexity of full attention constitutes a fundamental throughput bottleneck for long-context Transformers is not necessarily a principled expressiveness limitation of Transformer-based architectures."

---

> > ### Comment · Reviewer_VMLy · 2025-11-25
> >
> > Thanks for the clarifications. I maintain my high score.

---

### Official Review · Reviewer_iCy5 · 2025-10-31

**Soundness:** 3
**Presentation:** 3
**Contribution:** 2
**Rating:** 4
**Confidence:** 3

**Summary:**

This paper contains a recent line of work providing Turing-completeness proofs for Transformers. This paper specifically provides a construction of constant bit-size (that is, constant number of parameters at constant precision) transformers with bounded (moving) context window and custom relative positional encoding vectors that simulates Turing machine computations with a smaller slowdown than in the closest prior work (Li&Wang 2025). That prior construction used \Omega(s(n)) CoT steps to simulate a single step of a Turing machine operating in space s(n). In contrast, the current paper cuts this down to O(s(n)^c) for arbitrarily small positive c, where larger transformers can achieve smaller c.

**Strengths:**

- Contributes to emerging theoretical understanding of the Turing completeness of Transformers.
- Improves over prior work by reducing the CoT overhead in a constant bit-size setting.

**Weaknesses:**

- The design of the model is nonstandard. In particular, adding relative positional encoding vectors in line 195 appears confusing: is the idea that the encoding vector pos(i-j) depends on both the current position j and a later position i from which an attention head looks back at position j? This seems to make both parallel training impossible and autoregressive decoding extremely inefficient, as the whole transformer activations would have to be recomputed throughout the entire context for every next-token generation?

- The construction assumes positional encodings depending on the space bound s(n), and thus does not inherently length-generalize over all n. Is it really fair then to say that the model has fixed bit-size?

Missing references:
- line 077: the claim about practical attention patterns appears speculative, especially the link to the geometric progression attention. Is there a reference for the claim?
- line 080: can the authors provide a citation for this "common argument"?

**Questions:**

Missing citations:
- Another relevant recent paper is [1], which provides another Turing completeness construction, and also proves optimality of CoT lengths.

[1] Amiri et al, Lower Bounds for Chain-of-Thought Reasoning in Hard-Attention Transformers, ICML 2025

---

> ### Author Response · Authors · 2025-11-21
>
> We greatly value the reviewer's suggestions and feedbacks.
>
> **Q1:** The design of the model...as the whole transformer activations would have to be recomputed throughout the entire context for every next-token generation?
>
> **A1:** We sincerely thank the reviewer for pointing out this modeling issue, which we indeed missed in the original version.
>
> In the revised version (Sec. 2.3), we now introduce relative positional information **only inside the self-attention scores**, following the standard formulation of [1]. Concretely, we compute the attention score as
> $$
> s_{k,i}^\ell=\mathrm{hardmax}\bigl(
> \langle h_1^\ell K_k^\ell + \mathrm{pos}(i-1), h_i^\ell Q_k^\ell\rangle,\dots,
> \langle h_i^\ell K_k^\ell + \mathrm{pos}(i-i), h_i^\ell Q_k^\ell\rangle
> \bigr)
> $$
>  This avoids the concern about parallel training and autoregressive decoding: it does not require to recompute past transformer activations. We have also updated the proof of Theorem 3 so that the Transformer construction works with this type of relative PE.
>
> We remark that our conclusions (specifically Theorems 1 and 3) continue to hold under other relative positional encoding formations, e. g., using $\langle h^\ell_j\cdot K_{k}^{\ell},h^\ell_i\cdot Q_k^{\ell} \rangle+\mathrm{pos}(i-j)$ instead of $\langle h^\ell_j\cdot K_{k}^{\ell}+\mathrm{pos}(i-j),h^\ell_i\cdot Q_k^{\ell} \rangle$. In fact, our construction uses relative PE only to realize attention to **fixed offsets**.
>
>
>
> [1] Peter Shaw et al. Self-attention with relative position representation. NAACL 2018.
>
>
> **Q2:** The construction assumes positional encodings depending on the space bound s(n), and thus does not inherently length-generalize over all n. Is it really fair then to say that the model has fixed bit-size?
>
> **A2**: We appreciate the reviewer’s observation and agree that this aspect was not clearly explained in the original draft.
>
> - In our setting, “fixed bit-size” refers to the number and precision of **learnable parameters**. As can be checked from the proof of Theorem 3, we can simulate longer inputs by **keeping all learnable parameters unchanged and only adjusting the relative PE** (as defined in Equation (2)). We treat PE as part of the model architecture rather than as learnable parameters, so this adjustment does not affect the bit-size.
> - We briefly justify why the bit-size only counts learnable parameters. First, this matches how “model size” is reported in practice, where only trainable weights are counted. Second, it follows the standard separation in complexity theory between program description and resource usage: the learnable parameters correspond to a finite program, while the context window and PE represent available memory. Under this view, constant bit-size means we learn a single finite model that can operate on inputs of arbitrary length as long as sufficient memory is provided, just as a TM with a finite transition function can use an unbounded tape.
> - Moreover, even **if one counts the PE as part of the parameter budget (so that the bit-size is no longer constant), our conclusions still appear nontrivial**: it achieves optimal space efficiency (using an $O(s(n))$-long window), while significantly reducing the slowdown compared to prior $O(s(n))$-window simulations.
> - That said, we fully acknowledge that our relative PE design assumes that an upper bound on the space usage is known in advance. In the conclusion, we highlight as an open problem the design of PE that achieve the same simulation guarantees without assuming a known space bound in advance.
>
> In the revision, we added a remark at the end of Section 4 to include the above discussion.

---

> > ### Author Response · Authors · 2025-11-21
> >
> > **Q3:** line 077: the claim about practical attention patterns appears speculative, especially the link to the geometric progression attention. Is there a reference for the claim?
> >
> > **A3:** Thank you for pointing this out. Our intention was to make a qualitative, rather than quantitative, comparison to practice.
> >
> > *Empirical analyses:* Recent analyses of long-context LLMs show that attention matrices are typically **sparse and spatially structured**, with many heads focusing on local neighborhoods and a minority capturing global or long-range dependencies.
> >
> > - The sparsity of attention scores in pretrained LLMs, especially in long-context scenarios, has been well-documented, e.g. [2,3,4]. For example, as shown in [4], “some heads capture global dependencies while many focus mainly on local neighborhoods”.
> > - Moreover, [5] further identify three characteristic patterns of attention weights:
> >   - *A-shape heads*: whose weights concentrate on initial tokens and local windows;
> >   - *Vertical-Slash heads:* that attend to specific tokens and tokens at fixed offset;
> >   - *Block-Sparse heads:* exhibit spatial clustering.
> > - In addition, motivated by the "local-plus-sparse-global" patterns, several hybrid attention architectures alternate local and global attention. For examples, GPT-OSS alternates between full-attention (global) and local window attention (local).
> >
> > *In light of your comment*, we have softened the sentence in line 77 to avoid overstating the empirical link to geometric-offset attention. In the revised version we now say that our geometric-offset pattern is “in the same spirit” as these local-plus-sparse-global patterns observed in practice, and we add the above citations to support this qualitative analogy rather than claiming an exact match.
> >
> >
> >
> > [2] Xiao et al. Efficient streaming language models with attention sinks. ICLR 2024.
> >
> > [3] Ribar et al. Sparq attention: Bandwidth-efficient LLM inference. ICML 2024.
> >
> > [4] HSA: Head-wise Sparse Attention for Efficient and Accurate Long-Context Inference.
> >
> > [5] Jiang et al. MInference 1.0: Accelerating Pre-filling for Long-Context LLMs via Dynamic Sparse Attention. NeurIPS 2024.
> >
> >
> >
> > **Q4:** line 080: can the authors provide a citation for this "common argument"?
> >
> > **A4:** Rather than to introduce a new formal argument, our intention was to summarize the very broad concern about the quadratic time complexity of full attention, which poses a critical bottleneck for scaling transformers to long contexts.
> >
> > In the revision, we instead write that our results "suggests that the wide concern that the quadratic time complexity of full attention constitutes a fundamental throughput bottleneck for long-context Transformers (e.g., [6,7]) is not necessarily a principled expressiveness limitation of Transformer-based architectures."
> >
> >
> >
> > [6] Tay, et al. Efficient Transformers: A Survey. ACM Computing Surveys, 2022
> >
> > [7] Ruhi Sarikaya. Path to Artificial General Intelligence: Past, present, and future. Annual Reviews in Control, 2025.
> >
> >
> >
> > **Q5:** Missing citations: Another relevant recent paper is [1], which provides another Turing completeness construction, and also proves optimality of CoT lengths.
> >
> > **A5:** This reference is added in the revision.

---

> > > ### Comment · Reviewer_iCy5 · 2025-11-21
> > >
> > > Thanks. Have you already posed the revision PDF or are you going to do so soon?

---

> > > > ### Author Response · Authors · 2025-11-21
> > > >
> > > > The revision PDF has been posted. Thank you.

---

> > > > > ### Comment · Reviewer_iCy5 · 2025-11-22
> > > > >
> > > > > Thanks a lot.
> > > > >
> > > > > You've clearly addressed my concern about the positional encodings.
> > > > >
> > > > > And thanks for clarifying the sense in which the model has fixed bit size, and for acknowledging this:
> > > > >
> > > > > > That said, we fully acknowledge that our relative PE design assumes that an upper bound on the space usage is known in advance. In the conclusion, we highlight as an open problem the design of PE that achieve the same simulation guarantees without assuming a known space bound in advance.
> > > > >
> > > > > I will think about this more and let you know if I have questions.
> > > > >
> > > > > My original score indicates that I'm not opposed to accepting the paper, and that certainly remains true.

---

### Author Response · Authors · 2025-11-21
**One practically relevant implication of our theory**

We would like to highlight one practically relevant implication of Theorem 1. In the regime of practical interest—where the space complexity $s(n)$ is at most, say, $2^{500}$—the geometric offsets used by our Transformer construction in Theorem 1 reduce to the set $\{1,2,4,8,…\}$, truncated by the window size.

Concretely, the practical head–layer product $K$ is typically on the order of $10^3$-$10^4$; and it is lossless to focus on $2$-tape TMs, since $2$-tape TM can efficiently simulate multi-tape TMs with only a logarithmic slowdown.  So, if we take $K=6\times 10^3$ and $k=2$, then the common ratio in the geometric progression satisfies $\lceil s(n)^{6k/K}\rceil=2$ for any $s(n)\leq 2^{500}$.

Motivated by this, our theory points to the specific geometric offsets $\{1,2,4,8,…\}$ as a promising sparse attention pattern.

(We would like to especially thank Reviewer vfEe, whose comment about practical implementations of geometric-offset attention directly inspired this practically relevant implication.)

---

### Meta-Review · Area_Chair_hxUe · 2025-12-29

**Summary:**

This paper provides a theoretical contribution to the understanding of Transformer expressiveness. It specifically improves the efficiency bounds for simulating multi-tape Turing Machines (TMs) using constant bit-size Transformers. By leveraging synchronous multi-queue TMs as a bridge, the authors prove that a TM can be simulated with an optimal $O(s(n))$ context window and a significantly reduced chain-of-thought (CoT) overhead of $O(s(n)^c)$ for arbitrarily small $c > 0$.

The reviewers generally agreed that the work is mathematically rigorous and offers valuable insights, such as the proposed geometric-offset attention, which bridges theoretical universality with empirical practice. While initial concerns were raised regarding the clarity of definitions and the incremental nature of the work compared to prior results (e.g., Li & Wang 2025), the rebuttal effectively addressed these points by demonstrating a superior total CoT length reduction and providing clearer technical illustrations.

**Reviewer Concerns:**

Most concerns have been addressed during the rebuttal:

(1)	Reviewer iCy5 was concerned that the custom relative positional encoding (PE) would make parallel training and autoregressive decoding inefficient. The authors revised the model to use a standard relative PE formulation within self-attention scores, eliminating the need to recompute past activations.

(2)	Reviewer Carr noted that the initial comparison focused only on per-step overhead. The authors updated the analysis and Table 1 to show that their construction reduces the total CoT length.

(3)	Reviewer Carr identified a lack of explicit definition for TM space complexity. The authors clarified that and corrected the theorems accordingly.

(4)	Reviewers iCy5 and VMLy questioned whether the model was truly "fixed bit-size" if PE changed with $n$. The authors clarified that "fixed bit-size" refers to learnable parameters, while PE is treated as part of the architecture, similar to a TM's finite program using an unbounded tape.

(5)	Reviewer Carr questioned the novelty of synchronous queues relative to Hühne (1993). The authors explained that their synchrony requirement (every queue must pop/push every step) is more restrictive and requires new balance invariants and buffer mechanisms not present in the original work.

Though some concerns raised by reviewers haven’t been completely addressed:

(1)	The authors acknowledge that their relative PE design assumes an upper bound on space usage is known in advance. They have highlighted the design of PE that achieves simulation guarantees without this assumption as an open problem.

(2)	Reviewers suggested that empirical simulations would make the results more tangible. The authors have added this as a future direction but did not provide new experiments during the rebuttal period.

**Reviewer Scores:**

Reviewer iCy5: Originally 4 (Marginally below). If they had participated fully, they likely would have moved to a higher score, as they stated the authors "clearly addressed" their concerns and they were "not opposed to accepting the paper".

Reviewer VMLy: Originally 8 (Accept). This reviewer maintained their high score after the rebuttal.

Reviewer Carr: Originally 4 (Marginally below). Their score likely would have moved a little higher, as they noted that the authors' first-round responses "largely address" their concerns.

Reviewer vfEe: Originally 10 (Strong Accept). This reviewer remained highly positive about the manuscript.

---

### Decision · Program_Chairs · 2026-01-26

Accept (Poster)